# Effect of Dextrose Supplementation in the Pre-Ovulatory Sow Diet to Reduce Seasonal Influences on Litter Birth Weight Variation

**DOI:** 10.3390/ani9121009

**Published:** 2019-11-21

**Authors:** Kate Plush, Dannielle Glencorse, Jena Alexopoulos, Sally Tritton, Roy Kirkwood, Darryl D’Souza

**Affiliations:** 1SunPork Group, Murarrie, QLD 4172, Australia; dannielle.glencorse@sunporkfarms.com.au (D.G.); jena.alexopoulos@sunporkfarms.com.au (J.A.); sally.tritton@sunporkfarms.com.au (S.T.); darryl.dsouza@sunporkfarms.com.au (D.D.); 2School of Animal and Veterinary Sciences, The University of Adelaide, Roseworthy, SA 5371, Australia; roy.kirkwood@adelaide.edu.au

**Keywords:** dextrose, season, sow reproduction, piglet, weight variation, lactation growth

## Abstract

**Simple Summary:**

Seasonal infertility manifests in either pregnancy failure, or reduced litter size, or both in sows bred during summer. What is less clear is whether season influences piglet traits important for post-natal survival and growth, such as birth weight variation. Sows fed a diet with dextrose before ovulation gave birth to litters with reduced birth weight variation. Thus, this experiment had two aims: first, to identify if sows bred during summer would give birth to litters with higher birth weight variation, and second, to test if feeding sows dextrose before mating would reduce birth weight variation, especially during summer months. This experiment identified that there is evidence for increased birth weight variation in sows bred during the summer months despite the observed reduced litter size, suggesting that this is another way seasonal infertility can manifest. The 5% inclusion of dextrose in the pre-ovulatory sow diet increased litter size without compromising litter homogeneity at birth and resulted in higher piglet growth rates in those born to sows mated in winter (lactated in summer). Dextrose can be used successfully in a ‘wean to mate’ diet to boost the reproductive performance of sows.

**Abstract:**

In this experiment, we proposed two hypotheses: birth weight variation would increase in litters from sows bred in summer, and dextrose supplementation during the wean to mate period would ameliorate this manifestation of seasonal infertility. Five hundred and ninety-one multiparous sows were allocated to Control; standard diet, or Dextrose; control +5% dextrose diets from weaning until insemination during summer and winter. Dextrose sows farrowed 1.0 and 1.4 more total pigs born and pigs born alive than Control sows, respectively (*p* < 0.05). There was a tendency for a higher co-efficient of variation (CV) birth weight in summer than winter (16.6 ± 0.4 versus 15.8 ± 0.4; *p* = 0.1), but no effect of treatment or interaction between treatment and season was observed. Piglet average daily gain was unaffected in those born to sows bred in summer, but when born to sows bred in winter, Dextrose piglets grew 23 g per day faster than Control (*p* < 0.05). This experiment identified evidence for increased birth weight CV in sows bred during the summer months despite the reduced litter size, suggesting that this is another way seasonal infertility can manifest.

## 1. Introduction

Seasonal infertility can manifest itself in various forms but reports typically concentrate on a reduction in farrowing rate and litter size. This is driven by temperature and photoperiods, which directly impact on reproductive axes and indirectly act through effects on lactation feed intake. These effects can vary across years and are often site-dependent, as climatic conditions and management strategies differ. In herds that suffer reproductive loss during summer months, either through reduced farrowing rates or pigs born per litter or a combination of both, management strategies have been adopted to maintain pig flows. As a rule of thumb, producers will mate more sows during the warmer months of the year to counteract this dip in reproduction and maintain pig numbers on the ground. What producers may be less able to deal with is a dip in piglet birth weight, or, even more importantly, an increase in litter weight variation, as there are currently very few management strategies that can address such an issue. An increased within-litter heterogenicity is associated with increased piglet death and increased variation in weaning weights [1,2]. This last point may translate to impacts on time to slaughter, carcass weight variability and even meat quality [3]. Thus, effects on birth weight and birth weight variation are far-reaching across the production chain [4].

Campos, et al. [5] have suggested that higher levels of luteinising hormone (LH) will result in an increase in large follicle population, as they contain LH receptors whilst small follicles do not. They then go on to state that, if the follicle population is more uniform, so too will be the oocyte quality, and so conclude that high levels of LH will result in reduced litter heterogenicity. It has been proposed that reduced LH concentrations are a potential mechanism to explain seasonal infertility [6], with reduced levels reported during the summer months. Logically, if LH secretion is reduced in summer months, ovulation from smaller follicles may be more frequent, resulting in more variable oocyte quality. There is little published research investigating the effect of season on litter birth weight variability. Quesnel, et al. [7] reported no significant effect of season at conception on the coefficient of variation (CV) of birth weight (*p* = 0.068), and, because significance was not obtained, data was not presented. This study also failed to show any seasonal effect on litter size (conception rate was not reported), which may suggest the experimental herd was not susceptible to the impacts of seasonal infertility.

Birth weight CV appears to be established during the embryonic stage (d35 gestation) [8], suggesting that it is likely that the quality of the follicle and/or oocyte, rather than subsequent uterine conditions, contribute to eventual litter weight variance. To support this, when dietary arginine concentrations were increased to enhance placental growth from d30 gestation, no improvement in litter weight variation was reported [9]. Ovulation rate is positively related to birth weight CV [7], but one manifestation of seasonal infertility is a reduction in litter size and so there must be other metabolic influences acting to increase variation during summer breedings. Quesnel, et al. [7] discussed the impact of body reserve loss in lactation on birth weight heterogeneity in the subsequent litter, with the reasoning that metabolic status influenced follicle and oocyte quality and so embryo development. Heat stress during lactation has shown to reduce the circulating concentrations of both insulin and IGF-1 [10], both of which are strongly implicated in embryo quality. Carbohydrate-rich diets, fed during the follicular phase, appear to increase follicle and oocyte quality most likely explained by increases in plasma insulin and IGF-1. The inclusion of dextrose, in the diet from weaning to oestrus, has been shown to reduce birth weight CV of the litter from 21% to 17% [11]. There are several factors in this study that led the investigators to believe effects may be even greater than reported. Namely, authors state that only sows that had moderate body condition loss in lactation were used, and the effect of season was not investigated.

Variation in litter birth weight can affect pork production from the time of birth through to slaughter. There is little published information on the seasonality of variation in litter birth weight, but the science would suggest it is plausible. It would appear that oocyte quality is important, and so nutritional strategies should target this reproductive phase. Dextrose is successful in reducing litter weight variation, and so its effects should be tested during the summer months. The aim of the following experiment was to determine whether CV of birth weight was influenced by season, and to test the effectiveness of dextrose administration at alleviating these effects.

## 2. Materials and Methods

All procedures were carried out with approval from Primary Industries and Regions South Australia. The experimental periods were applied over two season replicates; summer, February–March and winter, August–September. Climate data for each of the experimental periods are shown in Table 1.

### 2.1. Dry Sow Management

Sow history prior being recruited to the experiment is outlined in Table 2. The sows were older (3.1 ± 0.0) and lighter (214.2 ± 5.1 kg) in summer than winter (parity 3.0 ± 0.0 and 216.8 ± 4.2 kg; *p* < 0.05). There was no difference in season or treatment for any other variables examined.

Over four replicates in summer, and six replicates in winter, sows were weaned in groups of 40 to partially slatted pens with a space allowance of 2 m^2^/sow. Sows were hand fed 3.5 kg of each of the control and treatment diets daily at 0700 h. Diet information is presented in Table 3. Sows in the Control and Dextrose received 0% and 5.5% dextrose (0 g/day and 193 g/day), respectively, as per Van den Brand, et al. [11].

From weaning, sows were moved daily at 0800 h from the large pen into detection mating areas in groups of five, where fence-line contact with two mature boars was used to detect oestrus. If oestrus was not detected, sows were moved back to the large group pens, but, when standing behaviour was observed, the sows were relocated into mating stations. Once in the mating stations, sows received two post-cervical artificial inseminations 24 h apart and were fed as outlined above. All sows were inseminated with a pooled (*n* = 3 to 5 boars) terminal sire dose. After the second insemination, sows were relocated into partially slatted gestation pens in groups of 50 and fed via electronic sow feeder at a space allowance of 1.8 m^2^/sow. Sows were allocated to gestation pens based on size and mating date and so pens contained both treatments, as well as commercial sows from outside the experiment. Return rate at 21 days of gestation was determined in the presence of a mature boar, and pregnancy rate confirmed via ultrasonography at approximately 28 days of gestation. All sows were fed a gestation diet (13.0 DE MJ/kg) at 2.1 kg/day in summer and 2.4 kg/day in winter.

### 2.2. Lactation Sow Management

Sows mated during the summer replicate farrowed during winter (June and July), and winter in summer (December and January). At approximately day 110 of gestation, sows were moved into farrowing accommodation and housed in farrowing crates (1.8 × 2.4 m). All sows were fed a lactation diet (14.2 DE MJ/kg) at 2.4 kg/day until farrowing, and then ad libitum to weaning. On the day of farrowing, the following measurements were recorded: total born (TB), born alive (BA), born dead (BD) and mummified foetuses. Piglets born both alive and dead were weighed individually, and those alive were tagged to allow individual identification. Fostering occurred once daily at 1300 h and all piglet movement involving tagged piglets was noted. Age and reason for piglet mortality was recorded, as well as piglet removal for ill thrift. Tagged piglets were weighed again on day 21 of lactation. Individual piglet weights on day 1 and day 21 were used to calculate the total litter weight, minimum and maximum piglet weight, standard deviation (SD) and co-efficient of variation (CV) weight, and the percentage of piglets within the litter weighing less than 1.1 kg (on D1; bottom quartile). Piglets were counted at weaning to give number of pigs weaned per sow (NPW).

### 2.3. Statistics

All data were analysed in SPSS v25 (IBM, Armonk, New York, NY, USA) and significance established at *p* < 0.05 and tendency at *p* = 0.1. Normally distributed data were analysed using a general linear mixed model, but generalized linear mixed models were applied to binary data (bred < 7 days, pregnancy rate, farrowing rate) using binary logistic regression, and to count data (all piglet mortalities) using Poisson regression. All analyses contained parity, season, treatment and the interaction between season and treatment as fixed effects, and mated week as the random term. The number of sows retained for these analyses are outlined in Table 4.

Separate analyses were conducted for 1764 Control and 2076 Dextrose piglets with the individual being the statistical unit. Linear regression was used for weights at day 1 and day 21, and binary logistic regression for survival to birth, fostering and weaning. The model included birth sow parity, birth litter size group (<12, 12–15 and <15), birth weight group (<1.1 kg, 1.1–1.6 kg and >1.6 kg), CV group (<15%, 15%–20%, 20%–25% and >25%), sex, season, treatment, and the interaction with main effects and treatment. Birth sow, foster sow and mated week were fit as random terms.

## 3. Results

No significant interaction between season and treatment was identified for any mating performance measure. Wean to service interval (WSI) was 1.5 days shorter in summer than winter (*p* < 0.01; Table 5) but was unaffected by treatment. These results were mirrored in percent sows bred within seven days. Pregnancy rate was improved by 22% from summer to winter (*p* < 0.001). Similarly, farrowing rate was improved by 25% from summer to winter (*p* < 0.001), and there was a tendency for a 5.9% improvement in Dextrose compared to Control sows (*p* = 0.1).

TB was higher in Dextrose than Control sows (*p* < 0.05; Table 6). Similarly, BA was higher in Dextrose sows (*p* < 0.001). There was no impact on BD, but mummified was similar between treatment sows in summer, and reduced in Dextrose sows when compared with Control during the winter matings (*p* < 0.01). Pre-foster deaths were reduced in sows mated during winter than summer (*p* < 0.05); no difference was observed in post-foster deaths, but piglet removal was less in the winter bred sows (*p* < 0.05). Piglet removal was also reduced in the Dextrose sows when compared with Control (*p* < 0.05). There was a tendency for NPW to be improved in sows mated in winter than summer (*p* = 0.1).

No significant interaction between season and treatment was identified for litter weight characteristics. A tendency for higher day 1 total litter weight was established in Dextrose sows when compared to Control (19.3 ± 0.4 versus 18.4 ± 0.4, respectively; *p* < 0.1), but no treatment effect was observed in minimum (0.86 ± 0.02 kg), maximum (1.88 ± 0.02 kg), or SD (0.31 ± 0.01). There was a tendency (*p* = 0.1) for day 1 CV to be higher in summer than winter (Figure 1). Total litter weight at day 21 was unaffected by season and treatment (52.7 ± 2.2 kg). Minimum (4.54 ± 0.10 kg versus 3.73 ± 0.13 kg) and maximum day 21 (8.02 ± 0.18 kg versus 7.41 ± 0.20 kg) piglet weights were higher in litters born to sows mated in summer than winter (*p* < 0.001). Both the SD and CV at day 21 was increased in litters born to Dextrose treated sows (Figure 1; *p* < 0.05). Season also affected the CV weight on day 21, with sows bred during winter having higher variation than those bred in summer (Figure 1; *p* < 0.05).

There was no impact of birth sow parity, litter size group, or CV group on piglet survival at birth, to fostering or to weaning (*p* < 0.05). More piglets from sows bred in summer survived to weaning (90.7%) than those bred in winter (82.8%; *p* < 0.001). Birth weight group significantly influenced survival during birth, to fostering and weaning (*p* < 0.001; Figure 2), with a lower proportion of those <1.1 kg surviving, and the highest survival observed at >1.63 kg, at all three timepoints.

There was a tendency (*p* < 0.1) for more Dextrose female piglets to survive the birth process than both females and males from Control sows. A trend for reduced survival until fostering in male Control piglets, compared with females from the same treatment, and both genders from Dextrose sows, was also established (*p* < 0.1). A total of 91.2% of females but 83.9% of males survived to weaning in the Control treatment (*p* < 0.01), but no sex effect was observed in Dextrose litters (86.9% and 86.2%, respectively; Figure 3).

Piglet average birth weight was influenced by litter size, birth weight group and gender (*p* < 0.01), but was unaffected by birth sow parity, CV group and treatment (*p* > 0.05; Table 7. Weight at day 21 and ADG was highest in piglets with the heaviest birth weight, and in those born to sows bred in summer (*p* < 0.001). A significant breeding season by treatment interaction was identified, whereby there was no treatment effect in piglets born to sows bred in summer, but in those bred during winter, the weaning weight and growth was improved in the Dextrose piglets (by 0.47 kg and 23 g, respectively; *p* < 0.05).

## 4. Discussion

We hypothesised that season would influence litter weight variability, and that when sows were fed a diet supplemented with dextrose before ovulation, this seasonal influence would be ameliorated. Our results support this hypothesis, in part. There was some evidence of increased litter weight variation in litters born to sows bred in summer, however the addition of dextrose failed to reduce this variation both across and within seasons.

The first objective of this investigation was to identify whether seasonal infertility impacted on piglet traits such as birth weight, and within-litter weight variation. The experimental site was appropriate for the investigation, as the traditional indicators of seasonal infertility were observed; farrowing rate was 25% lower and litter size reduced by 0.4 pigs during summer, in comparison to the winter matings. Whilst no difference in average birth weight was identified, there was some evidence that day one litter weight CV was increased in sows bred during summer, which is in support of our hypothesis. Reviewed by De Rensis, et al. [12], reduced lactation nutrient intake in summer alters both the basal and surge release of LH. Only large follicles contain LH receptors, thus, when LH levels are high, the growth of these large follicles is encouraged, resulting in a more uniform oocyte release. But when LH levels are reduced, both large and small follicles develop, resulting in a more heterogenous oocyte quality, and so, potentially, embryonic population. Our findings would suggest that birth weight heterogeneity is increased in sows bred during summer months and so is another way seasonal infertility can present itself. Interestingly, by weaning, it was the winter bred litters that exhibited the highest variation. These litters would have farrowed in summer, and with the experimental site utilising naturally ventilated farrowing sheds, it is likely the conditions the sow experienced during farrowing and lactation rather than at breeding that resulted in this finding. Indeed, the CV litter weight the piglet was born into had no impact on ADG, whilst there was a significant 40 g/day difference between season.

Interestingly, litter weight variation at birth had no impact on piglet survival and growth, which contrasts with previous work [1]. It is, however, the smaller piglets that are at a greater risk of mortality in litters with high variation [2]. We were unable to replicate these findings (data not shown), in that the survival of piglets <1.1 kg was low (54%–64%), but consistently low across all CV weight groups. Perhaps it is the management of low birth weight pigs on the experimental site that contributed to this result. Low birth weight pigs born into high CV litters are disadvantaged with regards to colostrum intake as they are often out-competed at the udder by larger conspecifics. Split suckling involves removing the largest piglets from the sow after farrowing when weight disparities are evident, and successfully increases the survival of small piglets, presumably through improved colostrum intake [13]. This practice is performed routinely on the experimental site, and so, whilst survival of the small piglets was low overall, split suckling may have acted to dilute any impacts of high litter weight CV. Irrespective of litter weight variation, low birth weight piglets (<1.1 kg) were significantly disadvantaged with regards to survival and growth, which is not a novel finding.

Dextrose appears to be a more effective energy source in dietary induced insulin enhancement than starch or fat [14]. This was a significant finding as insulin and IGF-1 are important regulators of ovarian function, and this likely explains the current improvement in litter size. Subsequent work by these authors showed that litter weight variation was reduced by ~4% when sows were supplemented with dextrose from weaning to breeding [11]. The present findings do not agree with this previous work, as CV litter weight was not reduced in the Dextrose treatment both across and within seasons. Thus, the objective to reduce the amount of within-litter variation through the inclusion of dextrose in a ‘wean to mate’ diet during summer months was not achieved. Quesnel, et al. [15] also failed to replicate these earlier findings when dextrose in the wean to mate period was combined with arginine from day 77 of gestation. However, dextrose supplementation did improve litter size, with total pigs born being 1.0 higher and pigs born alive 1.4 higher in this treatment compared to the control. Given the positive correlation between litter size and CV litter weight (r^2^ = 0.394, Milligan, et al. [2]), this increase in litter size should have been accompanied by an increase in birth weight variation, but this was not the case. Thus, dextrose supplementation fed from weaning to mating improves litters size whilst maintaining litter weight variation.

Perhaps one of the most interesting findings was that pigs conceived in winter and suckled in summer grew 23 g/day faster in Dextrose than Control litters, and as such were half a kilogram heavier at weaning. Lactation growth rates are suppressed during warmer months by poor sow feed intake, and this would be exacerbated under the natural ventilation farrowing shed conditions of this experiment. Given there was no improvement in any of the birth weight characteristics, this is a difficult finding to explain. We can only surmise that the size of effect of treatment on litter size across seasons altered mammary tissue activation in the winter bred, summer suckled Dextrose litters. Litter size is a main contributor to milk production, with more piglets ensuring adequate gland drainage and subsequent improvement in milk output [16]. In summer-bred sows, Dextrose improved piglets born alive by 0.4 piglets per litter but, in winter-bred sows, this improvement was increased to 1.6 piglets per litter. Thus, before the fostering event at 24 h post-farrowing, Dextrose sows bred in winter but suckled in summer nursed an extra 1.2 piglets. In the same review, Hurley [16] stated that the most important time for mammary development is in the first few days following farrowing. So, the improvement in litter size, which was largest in winter-bred, summer suckled sows from the Dextrose treatment may have resulted in better mammary activation, and so helped to alleviate poor piglet lactation growth rates in summer.

## 5. Conclusions

In conclusion, supplementing dextrose at a 5% inclusion rate to sows in the ‘wean to mate’ period increases litter size without increasing the CV of litter weight. An additional benefit of the dietary treatment was an improvement in suckling piglet growth rate during the warmer, summer months.

## Figures and Tables

**Figure 1 animals-09-01009-f001:**
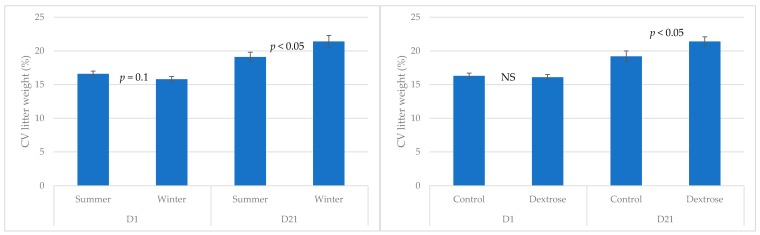
Co-efficient of variation (CV) litter weight (%) at birth (D1) and weaning (D21) for sows from Control and Dextrose treatments applied during the wean to mate period in summer and winter. *p*-value for season or treatment effect within measurement day are presented, with NS = not significant.

**Figure 2 animals-09-01009-f002:**
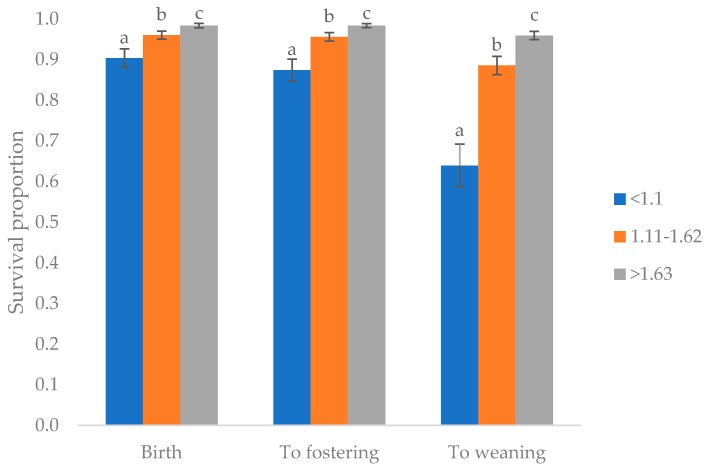
Proportion of piglets surviving birth, to fostering at 12–24 h of age, and to weaning at 21 days of age weighing <1.1 kg, 1.11–1.62 kg and >1.63 kg at birth. Superscripts denote significant difference at *p* < 0.001 within timepoint.

**Figure 3 animals-09-01009-f003:**
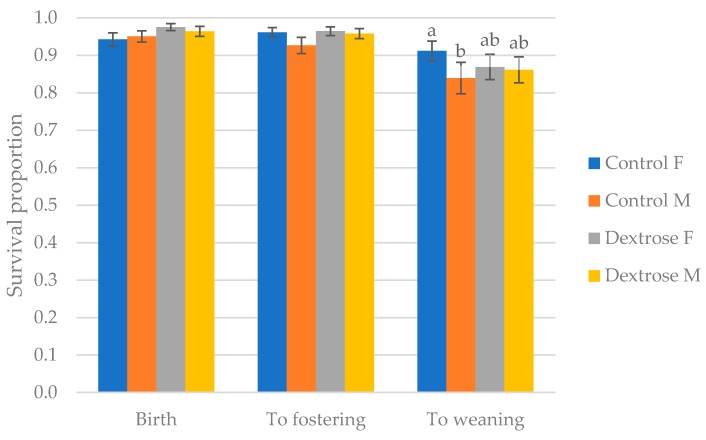
Proportion of piglets surviving birth, to fostering at 12–24 h of age, and to weaning at 21d of age, from Control and Dextrose treated sows of female (F) and male (M) gender. Superscripts denote significant difference at *p* < 0.01 within timepoint.

**Table 1 animals-09-01009-t001:** Climate data obtained from the Bureau of Meteorology weather station located in Roseworthy, approximately 22 km SE from the experimental site.

Season	Month	Min (°C)	Max (°C)	Highest Temperature (°C)	Days over 30 °C
Summer	February	16	32.5	43.3	18
	March	12.8	28.8	37.7	12
Winter	August	5.9	16.9	22.5	0
	September	3.6	19.6	28.6	0

**Table 2 animals-09-01009-t002:** Number and previous history of sows allocated to either the Control or Dextrose treatments during the summer and winter replicates. ^†^ * *p* < 0.05 with significant main effect presented in bracket.

Previous Measure	Summer				Winter				Sig ^†^
	Control		Dextrose		Control		Dextrose		
	Mean	SEM	Mean	SEM	Mean	SEM	Mean	SEM	
*n*	148		142		145		151		
Parity	3.1	0	3.1	0	3	0	3	0	* (Season)
Previous total pigs born (TB)	13.3	0.4	13.4	0.4	13.6	0.4	12.7	0.3	NS
Previous pigs born alive (BA)	12.5	0.3	12.7	0.4	12.6	0.3	11.9	0.3	NS
Previous pigs born dead (BD)	0.7	0.1	0.7	0.1	0.7	0.1	0.6	0.1	NS
PreviousMummified	0.6	0.1	0.5	0.1	0.7	0.1	0.6	0.1	NS
Previous number of pigs weaned (NPW)	9.5	0.2	9.8	0.2	9.8	0.2	9.7	0.2	NS
Back fat (mm)	16.5	0.3	16.3	0.3	17.1	0.3	17.5	0.3	NS
Weight (kg)	213.5	5.2	214.9	5.2	214	4.4	219.7	4.4	* (Season)

**Table 3 animals-09-01009-t003:** Diet information for Control and Dextrose treatments during the summer and winter replicates.

Diet Composition	Summer		Winter	
	Control	Dextrose	Control	Dextrose
Cereal (Barley/wheat) %	50.5	43.4	47.3	42.0
Digestible energy (DE) MJ/kg	13.4	13.4	13.4	13.4
Protein %	15.2	14.9	15.9	15.7
Lysine %	0.915	0.907	0.915	0.913
Crude fibre %	5.1	5.0	5.2	5.0
Dextrose %	0.0	5.5	0.0	5.5

**Table 4 animals-09-01009-t004:** Number of sows retained through each stage of the experiment.

Stage of Experiment	Summer		Winter	
	Control	Dextrose	Control	Dextrose
Allocated to treatment (*n*)	151	144	145	151
Mated in experiment (*n*)	129	122	98	107
Farrowing performance (*n*)	74	85	86	95
Lactation performance (*n*)	66	77	68	71

**Table 5 animals-09-01009-t005:** Mean ± SEM mating performance for sows from Control and Dextrose treatments applied during the wean to mate period in summer and winter. Superscripts denote significant difference at *p* < 0.05, with NS = not significant. ^†^ 95% confidence intervals rather than SEM are presented for binary data in brackets.

Breeding Performance	Season				Treatment				Season	Treatment
	Summer		Winter		Control		Dextrose			
	Mean	SEM	Mean	SEM	Mean	SEM	Mean	SEM		
WSI (days)	6.1 ^a^	0.4	7.6 ^b^	0.4	6.6	0.4	7.1	0.4	<0.01	NS
Bred < 7 days (%) ^†^	90.0 ^a^		74.0 ^b^		83.6		83.9		<0.001	NS
	(86.1–93.4)		(68–78.9)		(78.2–87.9)		(78.6–88.0)			
Pregnancy rate (%) ^†^	70.0 ^a^		92.0 ^b^		84.2		84.4		<0.001	NS
	(61.1–78.3)		(87.1–95.6)		(76.7–89.5)		(77.7–89.4)			
Farrowing rate (%) ^†^	64.0 ^a^		89.0 ^b^		75.7		81.6		<0.001	=0.1
	(56.2–70.3)		(83.5–92.6)		(68.4–81.8)		(75.3–86.6)			

**Table 6 animals-09-01009-t006:** Farrowing performance for sows from Control and Dextrose treatments applied during the wean to mate period in summer and winter. Superscripts denote significant difference at *p* < 0.05 within effect.

Farrowing Performance	Season				Treatment				Summer				Winter				*p*-Value		
	Summer		Winter		Control		Dextrose		Control		Dextrose		Control		Dextrose				
	Mean	SEM	Mean	SEM	Mean	SEM	Mean	SEM	Mean	SEM	Mean	SEM	Mean	SEM	Mean	SEM	Season	Treatment	Interaction
TB	13.8	0.3	14.2	0.3	13.6 ^a^	0.3	14.6 ^b^	0.3	13.6	0.4	14.1	0.4	13.7	0.4	14.8	0.4	NS	<0.05	NS
BA	12.5	0.3	13.1	0.2	12.3 ^a^	0.3	13.7 ^b^	0.3	12.3	0.4	12.7	0.4	12.3	0.4	13.9	0.3	<0.1	<0.001	=0.1
BD	1.0	0.1	0.8	0.1	0.9	0.1	0.8	0.1	0.9	0.1	1	0.1	0.9	0.1	0.7	0.1	NS	NS	NS
Mummified	0.8 ^a^	0.1	0.6 ^b^	0.1	0.7	0.1	0.7	0.1	1.0 ^a^	0.1	1.0 ^a^	0.1	0.7 ^b^	0.1	0.5 ^c^	0.1	<0.01	NS	<0.01
Pre-foster deaths	1.04 ^a^	0.09	0.83 ^b^	0.08	0.98	0.09	0.88	0.08	1.13	0.13	0.95	0.11	0.84	0.11	0.81	0.10	<0.05	NS	NS
Post-foster deaths	1.0	0.1	1.1	0.1	1.1	0.1	1.1	0.1	1.1	0.1	1.0	0.1	1.1	0.1	1.2	0.1	NS	NS	NS
Piglet removal	1.5 ^a^	0.1	1.1 ^b^	0.1	1.5 ^a^	0.1	1.2 ^b^	0.1	1.7	0.2	1.3	0.1	1.3	0.2	1.0	0.1	<0.05	<0.05	NS
NPW	9.0	0.2	9.4	0.2	9.1	0.2	9.3	0.2	8.8	0.2	9.2	0.2	9.4	0.2	9.4	0.2	=0.1	NS	NS

**Table 7 animals-09-01009-t007:** Mean (SEM) piglet birth weight (kg), day 21 weight (kg) and lactation average daily gain (ADG; g) for treatment and birth environment factors. Superscripts denote significant difference at p < 0.05, with NS = not significant. * denotes interaction between two main effects.

Factor	Birth Weight (kg)	Day 21 Weight (kg)	Lactation ADG (g)
Birth Sow Parity	NS	NS	NS
3	1.39 (0.01)	5.92 (0.15)	213 (6)
4	1.38 (0.01)	5.87 (0.15)	215 (6)
5	1.39 (0.01)	5.79 (0.15)	208 (6)
6	1.38 (0.01)	5.77 (0.22)	206 (9)
Litter Size	<0.001	NS	NS
<12	1.45 ^a^ (0.01)	5.95 (0.17)	214 (7)
12–15	1.37 ^b^ (0.01)	5.74 (0.13)	207 (6)
>15	1.34 ^c^ (0.01)	5.83 (0.15)	211 (6)
CV Group	NS	NS	NS
<15	1.40 (0.01)	5.73 (0.12)	202 (5)
15–20	1.39 (0.01)	5.82 (0.10)	208 (4)
21–25	1.36 (0.01	5.61 (0.17)	205 (7)
>26	1.41 (0.01	6.20 (0.38)	227 (16)
Birth Weight Quartile	<0.001	<0.001	<0.001
<1.1 kg	0.93 ^a^ (0.01)	4.67 ^a^ (0.15)	198 ^a^ (6)
1.11–1.62 kg	1.40 ^b^ (0.01)	5.93 ^b^ (0.13)	213 ^b^ (6)
>1.63 kg	1.84 ^c^ (0.01)	6.93 ^c^ (0.13)	220 ^b^ (6)
Gender	<0.01	NS	NS
Male	1.40 ^a^ (0.01)	5.83 (0.13)	210 (6)
Female	1.38 ^b^ (0.01)	5.85 (0.13)	211 (6)
Breeding Season	NS	<0.001	<0.001
Summer	1.39 (0.01)	6.13 ^a^ (0.14)	231 ^a^ (6)
Winter	1.39 (0.01)	5.55 ^b^ (0.15)	190 ^b^ (6)
Treatment	NS	NS	NS
Control	1.40 (0.01)	5.73 (0.18)	204 (8)
Dextrose	1.38 (0.01)	5.95 (0.16)	217 (7)
Breeding Season * Treatment	NS	<0.05	<0.05
Summer Control	1.40 (0.02)	6.13 ^a^ (0.19)	230 ^a^ (8)
Summer Dextrose	1.38 (0.01)	6.13 ^a^ (0.17)	231 ^a^ (7)
Winter Control	1.39 (0.02)	5.33 ^c^ (0.20)	179 ^c^ (9)
Winter Dextrose	1.39 (0.02)	5.80 ^b^ (0.19)	202 ^b^ (8)
Treatment * CV Group	NS	NS	NS
Treatment * Sex	NS	NS	NS
Treatment * Birth Weight Quartile	NS	NS	NS
Treatment * Litter Size	NS	NS	NS

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
