# Peer review of "Effect of Dextrose Supplementation in the Pre-Ovulatory Sow Diet to Reduce Seasonal Influences on Litter Birth Weight Variation"

_animals, 2019, doi:10.3390/ani9121009_

Round 1

Reviewer 1 Report

The author made responses to all my questions.

Author Response

The author made responses to all my questions.

We thank the reviewer for strengthening our manuscript

Reviewer 2 Report

During the first round of the review, authors addressed some of my reservations/suggestions but I still think that there is important room for improvement of the submitted manuscript. Please, find my remaining requests lower as the responses to your comments from the first round of the review.

In their current manuscript, Plush et al. aimed to study the effect of dextrose-supplemented diet on various parameters of the sows progeny. Their major hypotheses to test were whether sows bred during summer would give birth to litters with higher birth weight variation, and the second to test if feeding sows dextrose before mating would reduce birth weight variation, especially during summer months.

I think this study might be interesting for those closely related to the field of the pork production. The overall study design and statistical analysis is appropriate but authors should reorganize some data presentation aspects of the manuscript to point out the major conclusions made in their study.

The authors have removed the interaction from tables where no significance was established in order to make the tables easier to understand.

Since reviewer is not tightly connected to the field of the study I left on editors if the observed differences in the parameters and the and the overall impact of the study to the field of the pork production is worthy to be presented as research article in Animals.

Minor points:

I think that authors should more clearly identify in the individual parts of the manuscript what is the most interesting and novel finding they bring in the presented manuscript. Despite the fact that in the Simple summary they made to clear hypotheses to be tested, in the Introduction they present that the one of the hypothesis was already addressed in the literature.

The authors feel they have outlined how the current investigation differed from the previous study in which dextrose successfully reduced birth weight variation. ‘There are several factors in this study that lead the investigators to believe effects may be even greater than reported. Namely, authors state that only sows that had moderate body condition loss in lactation were used, and the effect of season was not investigated.

In the Results section the data  related to the tested hypotheses are presented in the very complex Table 7. At the end of discussion authors wrote: Perhaps one of the most interesting findings was that pigs conceived in winter and so suckled in summer grew 23g/day faster in Dextrose than Control litters, and as such were half a kilogram heavier at weaning. This finding is again not directly related to the hypotheses introduced at the beginning of the article and these facts decreasing the overall clarity of the manuscript.

The authors agree that the table was complex but were trying to follow how 2 x 2 factorial results are commonly presented. Given that the interaction between season and treatment was not significant for any of these measures, these columns have now been removed. The authors feel this makes the table easier to read. The finding of the improved growth rate was unexpected, and the authors agree that it was outside the hypotheses of the experiment. However, it is a very interesting finding and if removed from the discussion, would be missed by the reader. Historically, important scientific discoveries often fall outside the original experimental aims. We would like to suggest the paragraph relating to this improved growth is retained.

Response: I agree with the points written by the authors above, but for the clarity and logical flow of the manuscript, it would be worthy to clearly point out the results related to the tested hypotheses in the results section (my request about the data presentation lower) and in the Discussion part. Please, start your Discussion part clearly by the description of the outputs from your hypotheses testing – e.g. – our first hypothesis was – during the testing we conformed/not confirmed. The interesting findings out of the hypotheses scope can be of course presented, but it should be clearly stated in the discussion and abstract/summary (e.g. out of the scope of the hypotheses tested we observed interesting changes…).

I think that since almost all parameters related to the introduced hypotheses are present in the table 7, the data presentation of the parameters in this table should be expanded. When the parameters directly related to the tested hypotheses compared (like those mentioned in abstract) please provide the bar charts with the appropriate inferential statistical parameters. In the table it as also not very clear to which of the parameters (comparisons) p values are related and it should be again very well visible in the newly prepared charts presenting the most interesting findings.

Once again, we apologise for the complex table but given the design of the experiment (2 x 2 with season and diet) we feel this was the most appropriate way to present the results. As already explained, as the interaction was not significant was have removed all columns relating to these estimates, improving the readability of the table.

Response: I still insist that authors should present the most important results (with significant statistical differences) from the table 7 as charts. Please, prepare the charts as columns (box) charts showing the individual measurements as individual data points together with error bars and indicators of inferential statistical differences between individual groups. This will help reviewers and readers to make better picture about the data structure of your results and estimate better their importance/conclusiveness.

Furthermore, there are many data presented in the other figures which are not mentioned in the discussion at all (e.g. in figure 8 and others). If the authors are not discussing the presented data, those should be removed or go to the supplement.

This point is highly important since for example when authors concluding: In conclusion, supplementing dextrose at a 5% inclusion rate to sows in the ‘wean to mate’ period increases litter size without increasing litter weight variation it have to be clear that they are not using the comparison of SE parameters where litter size (n) affect the parameter of variance (SE).

The authors have added CV to the text to clarify how variation was measured.

I also think that another figures not directly related to the hypotheses tested can go to the Supplements.

Once again, we argue that whilst not pertinent to the hypotheses, they are still relevant findings, adding important discoveries in pig reproduction science.

Response: Please see my points above

Please increase the readability of the Table 7 by putting the SE parameters to one line.

Given the lack of significance for the interaction, these columns have now been removed and all estimates fit on a single line.

Author Response

In the Results section the data  related to the tested hypotheses are presented in the very complex Table 7. At the end of discussion authors wrote: Perhaps one of the most interesting findings was that pigs conceived in winter and so suckled in summer grew 23g/day faster in Dextrose than Control litters, and as such were half a kilogram heavier at weaning. This finding is again not directly related to the hypotheses introduced at the beginning of the article and these facts decreasing the overall clarity of the manuscript.

The authors agree that the table was complex but were trying to follow how 2 x 2 factorial results are commonly presented. Given that the interaction between season and treatment was not significant for any of these measures, these columns have now been removed. The authors feel this makes the table easier to read. The finding of the improved growth rate was unexpected, and the authors agree that it was outside the hypotheses of the experiment. However, it is a very interesting finding and if removed from the discussion, would be missed by the reader. Historically, important scientific discoveries often fall outside the original experimental aims. We would like to suggest the paragraph relating to this improved growth is retained.

Response: I agree with the points written by the authors above, but for the clarity and logical flow of the manuscript, it would be worthy to clearly point out the results related to the tested hypotheses in the results section (my request about the data presentation lower) and in the Discussion part. Please, start your Discussion part clearly by the description of the outputs from your hypotheses testing – e.g. – our first hypothesis was – during the testing we conformed/not confirmed. The interesting findings out of the hypotheses scope can be of course presented, but it should be clearly stated in the discussion and abstract/summary (e.g. out of the scope of the hypotheses tested we observed interesting changes…).

The discussion now begins with ‘’We hypothesised that season would influence litter weight variability, and that when sows were fed a diet supplemented with dextrose before ovulation, this seasonal influence would be ameliorated. Our results support this hypothesis, in part. There was some evidence of increased piglet weight variation in litters born to sows bred in summer, however the addition of dextrose failed to reduce this variation both across and within season.’’

I think that since almost all parameters related to the introduced hypotheses are present in the table 7, the data presentation of the parameters in this table should be expanded. When the parameters directly related to the tested hypotheses compared (like those mentioned in abstract) please provide the bar charts with the appropriate inferential statistical parameters. In the table it as also not very clear to which of the parameters (comparisons) p values are related and it should be again very well visible in the newly prepared charts presenting the most interesting findings.

Once again, we apologise for the complex table but given the design of the experiment (2 x 2 with season and diet) we feel this was the most appropriate way to present the results. As already explained, as the interaction was not significant was have removed all columns relating to these estimates, improving the readability of the table.

Response: I still insist that authors should present the most important results (with significant statistical differences) from the table 7 as charts. Please, prepare the charts as columns (box) charts showing the individual measurements as individual data points together with error bars and indicators of inferential statistical differences between individual groups. This will help reviewers and readers to make better picture about the data structure of your results and estimate better their importance/conclusiveness.

The authors have removed table 7 and replaced this with figures as per the reviewers request.

Furthermore, there are many data presented in the other figures which are not mentioned in the discussion at all (e.g. in figure 8 and others). If the authors are not discussing the presented data, those should be removed or go to the supplement.

Figure 2 (sex effects on piglet survival) has been removed from the manuscript. Is the reviewer referring to table 8 rather than figure 8? The results presented in this table are mentioned in the discussion (paragraph 4). 

Reviewer 3 Report

The authors have addressed my concerns.

Author Response

The authors have addressed my concerns.

We thank the reviewer for strengthening our manuscript

This manuscript is a resubmission of an earlier submission. The following is a list of the peer review reports and author responses from that submission.

Round 1

Reviewer 1 Report

In their current manuscript, Plush et al. aimed to study the effect of dextrose-supplemented diet on various parameters of the sows progeny. Their major hypotheses to test were whether sows bred during summer would give birth to litters with higher birth weight variation, and the second to test if feeding sows dextrose before mating would reduce birth weight variation, especially during summer months.

I think this study might be interesting for those closely related to the field of the pork production. The overall study design and statistical analysis is appropriate but authors should reorganize some data presentation aspects of the manuscript to point out the major conclusions made in their study.

Since reviewer is not tightly connected to the field of the study I left on editors if the observed differences in the parameters and the and the overall impact of the study to the field of the pork production is worthy to be presented as research article in Animals.

Minor points:

1) I think that authors should more clearly identify in the individual parts of the manuscript what is the most interesting and novel finding they bring in the presented manuscript. Despite the fact that in the Simple summary they made to clear hypotheses to be tested, in the Introduction they present that the one of the hypothesis was already addressed in the literature. In the Results section the data  related to the tested hypotheses are presented in the very complex Table 7. At the end of discussion authors wrote: Perhaps one of the most interesting findings was that pigs conceived in winter and so suckled in summer grew 23g/day faster in Dextrose than Control litters, and as such were half a kilogram heavier at weaning. This finding is again not directly related to the hypotheses introduced at the beginning of the article and these facts decreasing the overall clarity of the manuscript.

2) I think that since almost all parameters related to the introduced hypotheses are present in the table 7, the data presentation of the parameters in this table should be expanded. When the parameters directly related to the tested hypotheses compared (like those mentioned in abstract) please provide the bar charts with the appropriate inferential statistical parameters. In the table it as also not very clear to which of the parameters (comparisons) p values are related and it should be again very well visible in the newly prepared charts presenting the most interesting findings.

This point is highly important since for example when authors concluding: In conclusion, supplementing dextrose at a 5% inclusion rate to sows in the ‘wean to mate’ period increases litter size without increasing litter weight variation it have to be clear that they are not using the comparison of SE parameters where litter size (n) affect the parameter of variance (SE).

I also think that another figures not directly related to the hypotheses tested can go to the Supplements.

3) Please increase the readability of the Table 7 by putting the SE parameters to one line.

Author Response

In their current manuscript, Plush et al. aimed to study the effect of dextrose-supplemented diet on various parameters of the sows progeny. Their major hypotheses to test were whether sows bred during summer would give birth to litters with higher birth weight variation, and the second to test if feeding sows dextrose before mating would reduce birth weight variation, especially during summer months.

I think this study might be interesting for those closely related to the field of the pork production. The overall study design and statistical analysis is appropriate but authors should reorganize some data presentation aspects of the manuscript to point out the major conclusions made in their study.

The authors have removed the interaction from tables where no significance was established in order to make the tables easier to understand.

Since reviewer is not tightly connected to the field of the study I left on editors if the observed differences in the parameters and the and the overall impact of the study to the field of the pork production is worthy to be presented as research article in Animals.

Minor points:

I think that authors should more clearly identify in the individual parts of the manuscript what is the most interesting and novel finding they bring in the presented manuscript. Despite the fact that in the Simple summary they made to clear hypotheses to be tested, in the Introduction they present that the one of the hypothesis was already addressed in the literature.

The authors feel they have outlined how the current investigation differed from the previous study in which dextrose successfully reduced birth weight variation. ‘There are several factors in this study that lead the investigators to believe effects may be even greater than reported. Namely, authors state that only sows that had moderate body condition loss in lactation were used, and the effect of season was not investigated.

In the Results section the data  related to the tested hypotheses are presented in the very complex Table 7. At the end of discussion authors wrote: Perhaps one of the most interesting findings was that pigs conceived in winter and so suckled in summer grew 23g/day faster in Dextrose than Control litters, and as such were half a kilogram heavier at weaning. This finding is again not directly related to the hypotheses introduced at the beginning of the article and these facts decreasing the overall clarity of the manuscript.

The authors agree that the table was complex but were trying to follow how 2 x 2 factorial results are commonly presented. Given that the interaction between season and treatment was not significant for any of these measures, these columns have now been removed. The authors feel this makes the table easier to read. The finding of the improved growth rate was unexpected, and the authors agree that it was outside the hypotheses of the experiment. However, it is a very interesting finding and if removed from the discussion, would be missed by the reader. Historically, important scientific discoveries often fall outside the original experimental aims. We would like to suggest the paragraph relating to this improved growth is retained.

I think that since almost all parameters related to the introduced hypotheses are present in the table 7, the data presentation of the parameters in this table should be expanded. When the parameters directly related to the tested hypotheses compared (like those mentioned in abstract) please provide the bar charts with the appropriate inferential statistical parameters. In the table it as also not very clear to which of the parameters (comparisons) p values are related and it should be again very well visible in the newly prepared charts presenting the most interesting findings.

Once again, we apologise for the complex table but given the design of the experiment (2 x 2 with season and diet) we feel this was the most appropriate way to present the results. As already explained, as the interaction was not significant was have removed all columns relating to these estimates, improving the readability of the table.

This point is highly important since for example when authors concluding: In conclusion, supplementing dextrose at a 5% inclusion rate to sows in the ‘wean to mate’ period increases litter size without increasing litter weight variation it have to be clear that they are not using the comparison of SE parameters where litter size (n) affect the parameter of variance (SE).

The authors have added CV to the text to clarify how variation was measured.

I also think that another figures not directly related to the hypotheses tested can go to the Supplements.

Once again, we argue that whilst not pertinent to the hypotheses, they are still relevant findings, adding important discoveries in pig reproduction science.

Please increase the readability of the Table 7 by putting the SE parameters to one line.

Given the lack of significance for the interaction, these columns have now been removed and all estimates fit on a single line.

Reviewer 2 Report

Please clarify the dosage chosen reason for inclusion of 5% dextrose in the pre-ovulatory sow diet. Full name should be provided when the abbreviations were firstly appeared.

         Line 31 ……CV…….

Line 94-95, “Dextrose is successful in reducing litter weight variation”,reference(s) should added here. More information about the use of dextrose in sows should be added in the Introduction. Especially why the author considered that feeding sows dextrose before mating would reduce birth weight variation, especially during summer months, should be explained. The authors should carefully check whether those data in all Figures are significantly different (P < 0.01, P<0.001), since SEM seems pretty wide. Other composition of diet should be added in Table 3. Discussion need to be improved and the reason why dextrose supplementation fed from weaning to mating improves litters size should be added.

Author Response

Please clarify the dosage chosen reason for inclusion of 5% dextrose in the pre-ovulatory sow diet. Full name should be provided when the abbreviations were firstly appeared.

Citation added [11] in the methodology for reasoning behind 5%.

Line 31 ……CV…….

Definition ‘co-efficient of variation’ now added to abstract.

Line 94-95, “Dextrose is successful in reducing litter weight variation”,reference(s) should added here. More information about the use of dextrose in sows should be added in the Introduction. Especially why the author considered that feeding sows dextrose before mating would reduce birth weight variation, especially during summer months, should be explained.

The authors apologise that they were not specific enough in the paragraph preceding this statement. The text now reads ‘, and the inclusion of such a sugar, dextrose, in the diet from weaning to oestrus has been shown to reduce birth weight CV of the litter from 21% to 17% [11].’ We then follow on with the statement ‘There are several factors in this study that lead the investigators to believe effects may be even greater than reported. Namely, authors state that only sows that had moderate body condition loss in lactation were used, and the effect of season was not investigated’ which we feels addresses the second concern.

The authors should carefully check whether those data in all Figures are significantly different (P < 0.01, P<0.001), since SEM seems pretty wide.

The authors are confident that the P-values are correct and are happy to provide the reviewer with the SPSS output if requested.

Other composition of diet should be added in Table 3.

The authors have added the cereal grain component and lysine percentage of the diets to the table.

Discussion need to be improved and the reason why dextrose supplementation fed from weaning to mating improves litters size should be added. 

Without specific suggestions the authors are struggling how to incorporate the instruction that the discussion needs to be improved. Given this criticism was not made by the other two reviewers, no changes have been made to the discussion. How dextrose improves litter size in mentioned at the beginning of one of the paragraphs ‘Dextrose appears to be a more effective energy source in dietary induced insulin enhancement than starch or fat [14]. This was a significant finding as insulin and IGF-1 are important regulators of ovarian function.’ We have now added the following ‘and this likely explains the current improvement in litter size’.

Reviewer 3 Report

The manuscript is well structured and the topic is interesting. Nonetheless, revisions are needed in order for it to be publishable. The abstract is too long and, because of the high amount of numbers, quite hard to read. It should  be more direct and give the reader an overall idea of the background and outcomes. The introduction well covers the infertility topic, but does not introduce sufficiently dextrose and its potential usefulness; thus it is not clear why the authors decided to do the trial. Methods are clear, but my biggest concern regards the fact that in a reproductive study, few to none info are provided regarding the male counterpart. The quality of the semen influences the analyzed outcomes, so the author must provide clearer info regarding this section. Tables need to be better formatted. The discussion section is clear, but I think, again, that it should also take into account the potential biases of the study and the boars' effects. 

Author Response

The manuscript is well structured and the topic is interesting. Nonetheless, revisions are needed in order for it to be publishable. The abstract is too long and, because of the high amount of numbers, quite hard to read. It should  be more direct and give the reader an overall idea of the background and outcomes.

The abstract has been revised and now reads:

In this experiment we proposed two hypotheses: birth weight variation would increase in litters from sows bred in summer, and dextrose supplementation during the wean to mate period would ameliorate this manifestation of seasonal infertility. Five hundred and ninety-one multiparous sows were allocated to Control; standard diet, or Dextrose; control + 5% dextrose diets from weaning until insemination during summer and winter. Dextrose sows farrowed 1.0 and 1.4 piglets more total pigs born and pigs born alive than Control sows, respectively (P < 0.05). There was a tendency for a higher co-efficient of variation (CV) birth weight in summer than winter (16.6 ± 0.4 versus 15.8 ± 0.4; P = 0.1), but no effect of treatment or interaction between treatment and season was observed. Piglet average daily gain was unaffected in those born to sows bred in summer, but when born to sows bred in winter, Dextrose piglets grew 23g per day faster than Control (P < 0.05). This experiment identified evidence for increased birth weight CV in sows bred during the summer months despite the reduced litter size, suggesting that this is another way seasonal infertility can manifest.

The introduction well covers the infertility topic, but does not introduce sufficiently dextrose and its potential usefulness; thus it is not clear why the authors decided to do the trial.

The authors apologise. The section now reads:

Carbohydrate-rich diets fed during the follicular phase appear to increase follicle and oocyte quality most likely explained by increases in plasma insulin and IGF-1, and the inclusion of such a sugar, dextrose, in the diet from weaning to oestrus has been shown to reduce birth weight CV of the litter from 21% to 17% [11]. There are several factors in this study that lead the investigators to believe effects may be even greater than reported. Namely, authors state that only sows that had moderate body condition loss in lactation were used, and the effect of season was not investigated.

Methods are clear, but my biggest concern regards the fact that in a reproductive study, few to none info are provided regarding the male counterpart. The quality of the semen influences the analyzed outcomes, so the author must provide clearer info regarding this section.

For a publication on the ‘averaging effect’ of using pooled semen please see Foxcroft et al 2010. The following line has been added: All sows were inseminated with a pooled (n = 3 to 5 boars) terminal sire dose.

Tables need to be better formatted.

All tables in which the interaction between season and treatment was not significant have been simplified in order to improve readability.

The discussion section is clear, but I think, again, that it should also take into account the potential biases of the study and the boars' effects. 

We agree that this point would be valid if single sire matings had been carried out. However, we have now included a line in the methodology that we did indeed use pooled semen which was designed to remove significant boar factors on sow herds. We hope that clarifies the reviewer’s concerns.